# Lipid Emulsion Inhibits Amlodipine-Induced Nitric Oxide-Mediated Vasodilation in Isolated Rat Aorta

**DOI:** 10.3390/ijms24108741

**Published:** 2023-05-14

**Authors:** Kyeong-Eon Park, Soo Hee Lee, Sung Il Bae, Yeran Hwang, Seong-Ho Ok, Seung Hyun Ahn, Gyujin Sim, Soonghee Chung, Ju-Tae Sohn

**Affiliations:** 1Department of Anesthesiology and Pain Medicine, Gyeongsang National University Hospital, 15 Jinju-daero 816 Beon-gil, Jinju-si 52727, Gyeongsangnam-do, Republic of Koreasnugsoul@naver.com (S.I.B.); ash3832@gmail.com (S.H.A.);; 2Department of Anesthesiology and Pain Medicine, Gyeongsang National University Changwon Hospital, Changwon-si 51472, Gyeongsangnam-do, Republic of Koreamdoksh@naver.com (S.-H.O.); 3Department of Anesthesiology and Pain Medicine, Gyeongsang National University College of Medicine, Jinju-si 52727, Gyeongsangnam-do, Republic of Korea; 4Institute of Health Sciences, Gyeongsang National University, Jinju-si 52727, Gyeongsangnam-do, Republic of Korea; 5Department of Anesthesiology and Pain Medicine, Gyeongsang National University College of Medicine, Gyeongsang National University Hospital, 15 Jinju-daero 816 Beon-gil, Jinju-si 52727, Gyeongsangnam-do, Republic of Korea

**Keywords:** amlodipine, lipid emulsion, toxicity, vasodilation, nitric oxide

## Abstract

This study aimed to examine the effect of lipid emulsion on the vasodilation induced by a toxic dose of amlodipine in isolated rat aorta and elucidate its mechanism, with a particular focus on nitric oxide. The effects of endothelial denudation, N^W^-nitro-L-arginvine methyl ester (L-NAME), methylene blue, lipid emulsion, and linolenic acid on the amlodipine-induced vasodilation and amlodipine-induced cyclic guanosine monophosphate (cGMP) production were examined. Furthermore, the effects of lipid emulsion, amlodipine, and PP2, either alone or combined, on endothelial nitric oxide synthase (eNOS), caveolin-1, and Src-kinase phosphorylation were examined. Amlodipine-induced vasodilation was higher in endothelium-intact aorta than in endothelium-denuded aorta. L-NAME, methylene blue, lipid emulsion, and linolenic acid inhibited amlodipine-induced vasodilation and amlodipine-induced cGMP production in the endothelium-intact aorta. Lipid emulsion reversed the increased stimulatory eNOS (Ser1177) phosphorylation and decreased inhibitory eNOS (Thr495) phosphorylation induced via amlodipine. PP2 inhibited stimulatory eNOS, caveolin-1, and Src-kinase phosphorylation induced via amlodipine. Lipid emulsion inhibited amlodipine-induced endothelial intracellular calcium increase. These results suggest that lipid emulsion attenuated the vasodilation induced via amlodipine through inhibiting nitric oxide release in isolated rat aorta, which seems to be mediated via reversal of stimulatory eNOS (Ser1177) phosphorylation and inhibitory eNOS (Thr495) dephosphorylation, which are also induced via amlodipine.

## 1. Introduction

Lipid emulsion treats systemic toxicity resulting from local anesthetics [1]. In addition, lipid emulsion alleviates cardiovascular depression resulting from a highly lipid-soluble toxic dose of non-local anesthetic drugs, such as anti-depressants, anti-psychotics, and cardiovascular drugs, including verapamil and propranolol, which are unresponsive to supportive treatment [2,3]. Currently, the supportive treatments used for cardiovascular depression resulting from toxic doses of calcium channel blockers include gastrointestinal decontamination, fluid administration, correction of metabolic acidosis and electrolyte imbalance, vasopressors, calcium, high-dose insulin, and euglycemic therapy [4]. Lipid emulsion reportedly alleviates intractable cardiovascular depression resulting from a toxic dose of the antihypertensive drug amlodipine, which is a dihydropyridine L-type calcium channel blocker that acts on the vascular smooth muscles [4,5,6,7]. Furthermore, highly lipid-soluble racemic amlodipine (Log P: 3), which comprises S-amlodipine and R-amlodipine, is known to induce endothelial nitric oxide synthase (eNOS) phosphorylation and nitric oxide production, which may additionally contribute to amlodipine-induced vasodilation [8]. While racemic amlodipine and R-amlodipine induce nitric oxide production, S-amlodipine does not induce it [9]. Therefore, vasodilation evoked via amlodipine seems to be mediated mainly via the L-type calcium channel inhibition and partially via endothelial nitric oxide due to S- and R-amlodipine, respectively [8,9]. In addition, methylene blue, which is a non-specific inhibitor of guanylate cyclase (GC) in the cellular signal pathway associated with endothelial nitric oxide-induced vasodilation, attenuates the decreased blood pressure due to amlodipine toxicity in an in vivo rat experiment via inhibiting the nitric oxide-GC pathway-induced vasodilation [10,11]. Lipid emulsion also inhibits vasodilation induced using nitric oxide via inhibition of eNOS phosphorylation [12]. Furthermore, Intralipid increases blood pressure and vascular resistance but decreases flow-mediated vasodilation and vascular compliance in humans, which is possibly associated with decreased nitric oxide release [13,14]. However, the effect of lipid emulsion on amlodipine-induced nitric oxide-mediated vasodilation remains unknown. Therefore, in this study, we investigated the biological hypothesis that lipid emulsion attenuates amlodipine-induced nitric oxide-mediated vasodilation through inhibiting nitric oxide release. This study aimed to examine the effect of lipid emulsion (Intralipid) on the nitric oxide-mediated vasodilation induced using a toxic dose of amlodipine and elucidate the underlying mechanism, with a particular focus on nitric oxide.

## 2. Results

### 2.1. Effects of Endothelial Denudation, N^W^-nitro-L-arginine Methyl Ester, Methylene Blue, 1H-[1,2,4]oxadiazolo[4,3-a]quinoxalin-1-one, Calmidazolium, Lipid Emulsion, and Linolenic Acid, Either Alone or Combined, on the Amlodipine-Induced Vasodilation in Isolated Rat Aorta

Amlodipine (3 × 10^−7^ M)-induced vasodilation was higher in endothelium-intact than in endothelium-denuded aorta (Figure 1, *p* < 0.001 at 10–80 min). Dimethyl sulfoxide (DMSO, 0.1%), which is the same concentration used to dissolve the amlodipine, did not affect the phenylephrine-induced contraction in the endothelium-intact aorta (Appendix A). Nitric oxide synthase (NOS) inhibitor N^W^-nitro-L-arginine methyl ester (L-NAME, 10^−4^ M), non-specific GC inhibitor methylene blue (10^−6^ M), nitric oxide-sensitive GC inhibitor 1H-[1,2,4]oxadiazolo [4,3-a]quinoxalin-1-one (ODQ, 10^−5^ M), and calmodulin-regulated enzyme inhibitor calmidazolium (3 × 10^−5^ M) inhibited the amlodipine (3 × 10^−7^ M)-induced vasodilation (Figure 2, *p* < 0.01 vs. control at 10 min, and *p* < 0.001 vs. control at 20–80 min) in the endothelium-intact aorta. The post-treatment with lipid emulsion inhibited the amlodipine (3 × 10^−7^ M)-induced vasodilation (Figure 3A, *p* < 0.001 vs. control at 40–80 min) in the endothelium-intact aorta. However, it initially transiently increased the amlodipine-induced vasodilation (Figure 3A, *p* < 0.001 vs. control at 10 min). In addition, the lipid emulsion also inhibited the amlodipine (3 × 10^−7^ M)-induced vasodilation in endothelium-intact aorta pre-treated with L-NAME (Figure 3B, *p* < 0.05 vs. control at 10 min, and *p* < 0.001 vs. control at 20–80 min). However, the difference in the area (min%) under phenylephrine-induced contraction (%) time (30–80 min) curves between the groups with and without lipid emulsion treatment in the endothelium-intact aorta was greater without L-NAME (10^−4^ M) than with it (Figure 3C, *p* = 0.010). Linolenic acid (3 × 10^−6^ and 10^−5^ M) inhibited the amlodipine (3 × 10^−7^ M)-induced vasodilation in the endothelium-intact aorta (Figure 4A, 10^−5^ M: *p* < 0.01 vs. control at 10–80 min; 3 × 10^−6^ M: *p* < 0.05 vs. control at 30–80 min). However, linolenic acid (3 × 10^−6^ and 10^−5^ M) did not affect the vasodilation in the endothelium-denuded aorta (Figure 4B).

### 2.2. Effects of Lipid Emulsion, N^W^-nitro-L-arginine Methyl Ester, and Linolenic Acid, Either Alone or Combined, on Amlodipine-Induced Cyclic Guanosine Monophosphate Formation in Isolated Endothelium-Intact Rat Aorta

Amlodipine (3 × 10^−7^ M) increased cyclic guanosine monophosphate (cGMP) formation in the isolated endothelium-intact rat aorta (Figure 5A, *p* < 0.01 vs. control). However, lipid emulsion (1%) or L-NAME (10^−4^ M) inhibited the amlodipine (3 × 10^−7^ M)-induced cGMP formation (Figure 5A and B, *p* < 0.01 vs. amlodipine alone). In addition, linolenic acid (10^−5^ M) inhibited amlodipine (3 × 10^−7^ M)-induced cGMP formation (Figure 5C, *p* < 0.05 vs. amlodipine). However, the lipid emulsion (1%) did not affect the amlodipine (3 × 10^−7^ M)-induced cGMP formation in the endothelium-intact rat aorta pre-treated with L-NAME (10^−4^ M) (Figure 5B).

### 2.3. Effect of Lipid Emulsion and PP2 on the Endothelial Nitric Oxide Synthase, Caveolin-1, and Src-Kinase Phosphorylation Induced via Amlodipine in Human Umbilical Vein Endothelial Cells

Amlodipine (3 × 10^−7^ M) increased stimulatory eNOS (Ser1177) phosphorylation (Figure 6A, *p* < 0.05 vs. control) and decreased inhibitory eNOS (Thr 495) phosphorylation (Figure 6B, *p* < 0.001 vs. control) in human umbilical vein endothelial cells (HUVECs). However, lipid emulsion (1%) reversed the increased stimulatory eNOS (Ser1177) phosphorylation and decreased inhibitory eNOS (Thr495) phosphorylation induced using amlodipine (3 × 10^−7^ M) (Figure 6A,B, *p* < 0.001 vs. amlodipine alone). In addition, Src-kinase inhibitor PP2 (2 × 10^−5^ M) attenuated the amlodipine (3 × 10^−7^ M)-induced stimulatory eNOS (Ser1177) phosphorylation (Figure 6A, *p* < 0.01 vs. amlodipine alone). Furthermore, amlodipine (3 × 10^−7^ M) induced caveolin-1 (Tyr14) phosphorylation (Figure 7A, *p* < 0.01 vs. control) in HUVECs. However, Src-kinase inhibitor PP2 inhibited the amlodipine (3 × 10^−7^ M)-induced caveolin-1 (Tyr14) phosphorylation (Figure 7A, *p* < 0.001 vs. amlodipine alone). Amlodipine (3 × 10^−7^ M) also induced Src-kinase (Tyr 416) phosphorylation (Figure 7B, *p* < 0.001 vs. control) in HUVECs. However, PP2 inhibited the amlodipine (3 × 10^−7^ M)-induced Src-kinase (Tyr416) phosphorylation (Figure 7B, *p* < 0.001 vs. amlodipine alone).

### 2.4. Effects of Lipid Emulsion and N^W^-nitro-L-arginine Methyl Ester on the Intracellular Calcium Level Induced via Amlodipine in Human Umbilical Vein Endothelial Cells

Amlodipine (10^−6^ M) increased the intracellular calcium (Figure 8, *p* < 0.001 vs. control) in HUVECs. However, lipid emulsion (1%) and L-NAME (10^−4^ M) inhibited the calcium level increase via amlodipine (10^−6^ M) (Figure 8, *p* < 0.001 vs. amlodipine alone).

## 3. Discussion

In this study, we aimed to examine the effect of lipid emulsion on nitric oxide-mediated vasodilation induced via a toxic dose of amlodipine and elucidate the underlying mechanism. The results of our study suggest that lipid emulsion inhibits the nitric oxide-mediated vasodilation induced via amlodipine in endothelium-intact rat aorta via inhibiting endothelial nitric oxide release. The major findings of our study can be summarized as follows: (1) inhibitors of eNOS-nitric oxide-GC pathway attenuated amlodipine-induced vasodilation of the endothelium-intact rat aorta; (2) lipid emulsion and linolenic acid inhibited amlodipine-induced vasodilation in the endothelium-intact rat aorta; (3) lipid emulsion did not affect the amlodipine-induced cGMP formation in L-NAME-pretreated endothelium-intact aorta; (4) lipid emulsion reversed the increased stimulatory eNOS (Ser1177) phosphorylation and decreased inhibitory eNOS (Thr495) phosphorylation induced via amlodipine; and (5) L-NAME and lipid emulsion inhibited the increased intracellular calcium induced via amlodipine.

Endothelial nitric oxide is produced from L-arginine using eNOS, which is calcium-dependent and binds to calmodulin [15]. Endothelial nitric oxide activates GC, leading to cGMP formation [15]. The increased cGMP formation produces vasodilation through activating cGMP-dependent protein kinase [15]. Amlodipine-induced vasodilation was partially inhibited via the NOS inhibitor L-NAME, the non-specific GC inhibitor methylene blue, the nitric oxide-sensitive GC inhibitor ODQ, and the calmodulin-regulated enzyme inhibitor calmidazolium. Thus, these results suggest that vasodilation induced via racemic amlodipine (including S- and R-amlodipine) is partially mediated via a pathway involving calmodulin-dependent NOS-nitric oxide-GC. Endothelial nitric oxide is produced using racemic and R-amlodipine; however, it is not produced using S-amlodipine [9]. Thus, the nitric oxide-mediated vasodilation induced via racemic amlodipine in this study seems to be due to R-amlodipine. Lipid emulsion inhibits the nitric oxide-mediated vasodilation induced using acetylcholine and calcium ionophore A23187, possibly due to eNOS inhibition [12]. In addition, lipid emulsion increases left ventricular systolic pressure via inhibiting nitric oxide release [16]. Furthermore, lipid emulsion inhibits vasodilation evoked using the ATP-sensitive potassium channel agonist levcromakalim via inhibiting nitric oxide [17]. These reports suggest that lipid emulsion attenuates endothelial nitric oxide release. Consistent with the amlodipine-induced vasodilation inhibition mediated via lipid emulsion in endothelium-intact aorta pre-treated with L-NAME (Figure 3B), lipid emulsion inhibited toxic doses of amlodipine-induced vasodilation in the endothelium-denuded rat aorta via an indirect mechanism associated with lipid emulsion-mediated absorption of highly lipid-soluble amlodipine (Log P: 3) [18]. The magnitude of lipid emulsion-mediated inhibition of amlodipine-induced vasodilation was higher in the endothelium-intact rat aorta without L-NAME than that pre-treated with L-NAME (Figure 3C). Considering these previous reports, the lipid emulsion-mediated increased inhibition of the vasodilation induced using amlodipine’s toxic dose in the endothelium-intact aorta without L-NAME seems to be due to lipid emulsion-mediated inhibition of endothelial nitric oxide release using amlodipine’s toxic dose (therapeutic plasma concentration of amlodipine: approximately 3.1 × 10^−8^ M) [12,16,17,18,19]. Lipid emulsion inhibited early amlodipine-induced vasodilation at 10 min in the endothelium-intact aorta treated with L-NAME (Figure 3B). Therefore, the lipid emulsion-induced transient increase in amlodipine-induced vasodilation in the endothelium-intact aorta without L-NAME at 10 min (Figure 3A) may be associated with the transient activation of nitric oxide release. Linolenic acid increases phenylephrine-induced contraction and decreases acetylcholine-induced nitric oxide-mediated vasodilation, mediated via nitric oxide inhibition [20,21]. Linolenic acid inhibited vasodilation induced using amlodipine’s toxic dose in the endothelium-intact rat aorta. However, it did not affect amlodipine-induced vasodilation in the endothelium-denuded rat aorta. Consistent with previous reports, linolenic acid-mediated inhibition of vasodilation induced via amlodipine’s toxic dose seems to be mediated via linolenic acid-mediated inhibition of endothelial nitric oxide through inhibiting eNOS, leading to the attenuation of vasodilation [20,21]. Thus, lipid emulsion-mediated inhibition of nitric oxide-mediated vasodilation induced via a toxic amlodipine dose may be due to linolenic acid contained in Intralipid. Furthermore, the non-specific GC inhibitor methylene blue attenuated the decreased blood pressure induced using amlodipine’s toxic dose [10,11]. This attenuation suggests that improved blood pressure resulting from methylene blue is mediated through inhibiting the vasodilation induced via amlodipine-induced nitric oxide and GC pathway activation. However, lipid emulsion (1%) slightly reduced amlodipine’s concentration via sequestration [18]. Thus, clearly differentiating lipid emulsion’s direct inhibition of amlodipine-meditated stimulatory eNOS activation from lipid emulsion-mediated amlodipine absorption (the indirect mechanism) is very difficult, contributing to lipid emulsion-mediated inhibition of nitric oxide-mediated vasodilation induced using amlodipine’s toxic dose [18].

Consistent with inhibition of amlodipine’s toxic dose-induced vasodilation via lipid emulsion, linolenic acid, and L-NAME in the endothelium-intact aorta, L-NAME, linolenic acid, and lipid emulsion inhibited amlodipine-induced cGMP production in the endothelium-intact aorta (Figure 5). In contrast, lipid emulsion did not affect amlodipine’s toxic dose-induced cGMP formation in endothelium-intact aorta pre-treated with L-NAME (Figure 5B). This finding suggests that lipid emulsion-mediated inhibition of amlodipine’s toxic dose-induced cGMP formation is mediated via nitric oxide inhibition.

Endothelial nitric oxide production is increased through stimulatory eNOS (Ser1177) phosphorylation and inhibitory eNOS (Thr495) dephosphorylation [22]. Amlodipine in endothelial cells increases stimulatory eNOS (Ser1177) phosphorylation and reduces inhibitory eNOS (Thr495) phosphorylation, leading to nitric oxide formation and increased cGMP [8]. Consistent with a previous report, amlodipine increased stimulatory eNOS (Sre1177) phosphorylation and reduced inhibitory eNOS (Thr495) phosphorylation, contributing to endothelial nitric oxide production [8]. Lipid emulsion treatment reversed the increased stimulatory eNOS (Ser1177) phosphorylation and decreased inhibitory eNOS phosphorylation induced using amlodipine’s toxic dose, which may be associated with lipid emulsion-mediated inhibition of amlodipine-induced endothelial nitric oxide-dependent vasodilation. Caveolin mediates cellular signal transduction involving upstream signal components, such as receptor tyrosine kinase and G protein-coupled receptor, and downstream signal components, such as nitric oxide and ion channel, through the caveolin scaffolding domain [23]. The binding of eNOS with the caveolin’s scaffolding domain reduces eNOS activity [23]. Stimulatory eNOS and caveolin-1 phosphorylation, induced via alpha-2 adrenoceptor agonist dexmedetomidine, was reportedly inhibited using Src-kinase inhibitor PP2 [24]. In addition, bupivacaine-induced inhibitory eNOS (Thr495) phosphorylation was mediated via Src-kinase and caveolin-1 pathway, attenuating the vasodilation [25]. Similar to previous reports, Src-kinase inhibitor PP2 inhibited stimulatory eNOS (Ser1177) and caveolin-1 (Tyr14) phosphorylation evoked via amlodipine and attenuated amlodipine-induced Src-kinase (Tyr416) phosphorylation [24]. Thus, considering previous reports, these results suggest that amlodipine-induced eNOS phosphorylation (Ser1177) is mediated via Src-kinase-induced phosphorylation of caveolin-1 [23]. Amlodipine-induced inhibition of protein kinase C phosphorylation was found to be associated with amlodipine-induced alternation of eNOS phosphorylation [8]. However, further studies are needed to examine the detailed upstream cellular signal pathway contributing to the lipid emulsion-mediated reversal of amlodipine-induced stimulatory eNOS (Ser1177) phosphorylation and inhibitory eNOS (Thr495) dephosphorylation. eNOS activity is calcium-dependent [15]. Endothelial receptor-mediated vasodilator bradykinin increased endothelial calcium [26]. Furthermore, L-NAME attenuated the increase in calcium induced through flow in human aortic endothelial cells [27]. Amlodipine increased endothelial calcium in HUVECs. However, similar to a previous report, lipid emulsion and L-NAME inhibited the increased calcium level due to amlodipine, suggesting that this inhibition may contribute to reduced nitric oxide production via decreased stimulatory eNOS (Ser1177) activity [27].

This study has a few limitations. Firstly, the peripheral vascular resistance, which contributes to blood pressure, is mainly determined via a small resistance arteriole. However, in this study, the rat aorta was regarded as a conduit vessel [28]. Secondly, for isometric tension measurement, we used isolated rat aorta. However, for the western blot, we used HUVECs. HUVECs have disadvantages, such as sex differences and some degree of heterogeneity [29]. Thirdly, an in vivo study is more appropriate to examine the effects of lipid emulsion on cardiovascular depression resulting from amlodipine’s toxic dose. However, our study was an in vitro study. Therefore, a further study comparing the effect of lipid emulsion and non-specific GC inhibitor methylene blue on hypotension induced using a toxic dose of amlodipine in an in vivo model is required to confirm this result. Despite these limitations, lipid emulsion-mediated inhibition of nitric oxide-mediated vasodilation induced via amlodipine may mitigate the severe hypotension produced using a toxic dose of amlodipine through inhibiting nitric oxide. In addition, a toxic dose of amlodipine may produce less decreased blood pressure in patients with a compromised endothelium than a normal endothelium.

In summary, lipid emulsion inhibited toxic doses of amlodipine-induced nitric oxide-mediated vasodilation, being mediated via inhibiting nitric oxide production through reversing stimulatory and inhibitory eNOS phosphorylation.

## 4. Materials and Methods

The Institutional Animal Care and Use Committee of Gyeongsang National University approved the experimental protocol (GNU-211217-R0106, 22 December 2021). We conducted the study following the guidelines of Animal Care and Use.

### 4.1. Preparation of Rat Aorta and Isometric Tension Measurement

A male Sprague–Dawley rat with a body weight of 220–300 g (Koatech, Pyeongtaek, Republic of Korea) was anesthetized with 100% carbon dioxide. The isolated rat aorta for assessing isometric tension was prepared, as previously described [20]. The rat’s thoracic cavity was opened, and the descending thoracic aorta was extracted from the thoracic cage. The extracted thoracic aorta was put in Krebs solution containing sodium chloride (118 mM), sodium bicarbonate (25 mM), glucose (11 mM), potassium chloride (4.7 mM), calcium chloride (2.4 mM), magnesium sulfate (1.2 mM), and monopotassium phosphate (1.2 mM). The connective tissue and fat surrounding the isolated rat aorta in the Krebs solution were removed under a microscope. The isolated descending thoracic aorta was cut into 2.5-mm-long segments. The endothelium of some aortas was peeled off via rolling the aorta backward and forward using two 25-gauge needles inserted into the aortic lumen. The isolated rat descending thoracic aorta was suspended into a Grass isometric transducer (FT-03, Grass Instrument, Quincy, MA, USA) of organ bath, which was maintained at 37 °C. According to a previous report, a baseline resting tension of 24.5 mN is maintained for 1.5 h to achieve a plateau [30]. During this process, the fresh Krebs solution replaced the pre-existing Krebs solution. The pH of the Krebs solution was maintained at 7.4 through supplying it with 95% oxygen and 5% carbon dioxide. The endothelial integrity of endothelium-intact rat aorta was examined as follows [20]. After phenylephrine (10^−7^ M) produced a sustained and stable contraction, acetylcholine (10^−5^ M) was added to the organ bath. The aorta with >85% of the acetylcholine-induced relaxation from the phenylephrine-induced contraction was considered an endothelium-intact aorta (Appendix A). The endothelial denudation was verified as follows [20]. After phenylephrine (10^−8^ M) produced a sustained and stable contraction of the aorta, acetylcholine (10^−5^ M) was added to the organ bath. An aorta with <15% acetylcholine-induced relaxation was considered endothelium-denuded aorta (Appendix A). The endothelium-intact and endothelium-denuded rat aortas were washed several times to restore baseline resting tension, and the following experimental protocols were performed.

### 4.2. Experimental Protocols

Firstly, the effect of endothelial denudation on toxic dose of amlodipine-induced vasodilation was examined to determine whether amlodipine-induced vasodilation is endothelium-dependent [19]. After phenylephrine (10^−6^ M) produced sustained and stable contraction of isolated rat aorta with or without the endothelium, amlodipine (3 × 10^−7^ M) was added to the organ bath to induce vasodilation in endothelium-intact and endothelium-denuded rat aorta. The induced vasodilation was monitored for 80 min after adding amlodipine. In addition, we examined the effect of 0.1% DMSO, which is used to dissolve amlodipine (3 × 10^−7^ M), on the phenylephrine-induced contraction in the endothelium-intact aorta. After phenylephrine (10^−6^ M) produced a sustained and stable contraction, 0.1% DMSO was added to the organ bath. The phenylephrine-induced contraction was monitored for 80 min after adding DMSO.

Secondly, the cellular signal pathway involving NOS-nitric oxide-GC is associated with endothelial nitric oxide-induced vasodilation. Therefore, we evaluated the effects of its inhibitors on the amlodipine-induced vasodilation in endothelium-intact rat aorta to determine whether the vasodilation depends on the NOS-nitric oxide-GC pathway. Phenylephrine (10^−6^ M) produced a stable and sustained contraction after endothelium-intact rat aortas were pre-treated with the NOS inhibitor L-NAME (10^−4^ M), non-specific GC inhibitor methylene blue (10^−6^ M), nitric oxide-sensitive GC inhibitor ODQ (10^−6^ M), or calmodulin-regulated enzyme inhibitor calmidazolium (3 × 10^−6^ M) for 20 min [20]. Amlodipine (3 × 10^−7^ M) was added to the organ bath to induce vasodilation in the endothelium-intact rat aorta with or without inhibitors (L-MAME, methylene blue, ODQ, and calmidazolium). Thereafter, vasodilation was monitored for 80 min after adding amlodipine.

Thirdly, the effect of lipid emulsion (Intralipid) on the amlodipine-induced vasodilation in endothelium-intact rat aorta with or without L-NAME was investigated to determine whether lipid emulsion-mediated inhibition of amlodipine-induced vasodilation of the endothelium-intact rat aorta was endothelial nitric oxide-dependent. After some endothelium-intact rat aortas were pre-treated with NOS inhibitor L-NAME (10^−4^ M) for 20 min, phenylephrine (10^−6^ M) produced a stable and sustained contraction with or without L-NAME. Subsequently, amlodipine (3 × 10^−7^ M) was added to the organ bath, and some aortas were immediately post-treated with lipid emulsion (1%). Amlodipine-induced vasodilation was monitored for 80 min.

Fourthly, the effect of linolenic acid, which is a long-chain fatty acid in Intralipid, on amlodipine-induced vasodilation was investigated to determine whether linolenic acid-mediated inhibition of amlodipine-induced vasodilation is endothelium-dependent. After phenylephrine (10^−6^ M) produced a sustained and stable contraction in isolated rat aorta with or without the endothelium, amlodipine (3 × 10^−7^ M) was added to the organ bath, and some rat aortas were immediately post-treated with linolenic acid (3 × 10^−6^ and 10^−5^ M). Amlodipine-induced vasodilation was monitored for 80 min after adding amlodipine in the endothelium-intact or endothelium-denuded rat aorta, with or without linolenic acid.

### 4.3. Cyclic Guanosine Monophosphate (cGMP)

cGMP was measured in isolated endothelium-intact rat aorta, as described previously [24]. cGMP was measured using a cGMP complete kit (Abcam, Cambridge Science Park, Cambridge, England). The descending thoracic aorta with endothelium was placed into Krebs solution in a 10 mL organ bath for 60 min at 37 °C, including the drug treatment time. The endothelium-intact aortic strips were treated with amlodipine (3 × 10^−7^ M) alone for 5 min; lipid emulsion (Intralipid, 1%) alone for 25 min; lipid emulsion (1%) or linolenic acid (10^−5^ M) for 20 min, followed by amlodipine (3 × 10^−7^ M) for 5 min; L-NAME (10^−4^ M) for 30 min, followed by amlodipine (3 × 10^−7^ M) for 5 min; L-NAME (10^−4^ M) for 15 min, followed by lipid emulsion (1%) for 15 min and post-treatment with amlodipine (3 × 10^−7^ M) for 5 min. The descending thoracic aortic strips were then frozen with liquid nitrogen and homogenized in 10^−1^ M hydrochloride. Acidic supernatants were acetylated, and cGMP was measured using ELISA with the cGMP complete kit. The cGMP concentration from each aortic strip was expressed as ρmol/mL.

### 4.4. Human Umbilical Vein Endothelial Cells (HUVECs) Culture

HUVECs (C-003-5C, American Type Culture Collection, Manassas, VA, USA) were maintained in the endothelial cell medium (ECM) (ScienCell, Carlsbad, CA, USA) containing 15% fetal bovine serum (ScienCell), 1% endothelial cell growth supplement (Sciencell), 100 units/mL penicillin, and 100 µg/mL streptomycin (ScienCell). The cells were grown in a humidified atmosphere at 37 °C in a 5% CO_2_ incubator. The cells at passages 3–5 were further incubated for 4 h in serum-free ECM before undergoing drug treatment

### 4.5. Western Blot Analysis

The expression of eNOS (Ser1177 and Thr495), Src-kinase (Tyr416), and caveolin-1 (Tyr14) phosphorylation in HUVECs was assessed using western blot, as previously described [24]. Cells were treated with amlodipine alone for 10 min and 1% Intralipid for 1 h, followed by amlodipine for 10 min and PP2 for 30 min, followed by amlodipine for 10 min and either 1% lipid emulsion alone for 70 min or PP2 alone for 40 min to determine the expression of stimulatory eNOS (Ser1177) phosphorylation. For determining the expression of inhibitory eNOS (Thr495) phosphorylation, cells were treated with amlodipine alone for 1 min and 1% lipid emulsion for 1 h, followed by amlodipine for 1 min or 1% lipid emulsion alone for 61 min. Cells were treated with amlodipine alone for 5 min and PP2 for 30 min, followed by amlodipine for 5 min or PP2 alone for 35 min to determine the expression of Src-kinase (Tyr416) and caveolin-1 (Tyr14) phosphorylation. After treatment, the cells were harvested in a radio-immunoprecipitation assay buffer (Cell Signaling Technology, Beverly, MA, USA) with a protease inhibitor cocktail (Thermo Fisher Scientific, Rockfield, IL, USA) and a phosphatase inhibitor cocktail (Thermo Fisher Scientific). The lysates were centrifuged at 20,000× *g* for 15 min at 4 °C, and the protein content of the supernatant was quantified using a bicinchoninic acid protein assay reagent kit (Thermo Fisher Scientific). After 10 min of boiling, protein samples were separated using sodium dodecyl sulfate-polyacrylamide gel electrophoresis and transferred onto the polyvinylidene difluoride membrane (Millipore, Bedford, MA, USA). Membranes were blocked in 5% skimmed milk in tris-buffed saline with 0.5% Tween-20 (TBST) for 60 min at 25 °C and incubated with primary antibodies (anti-phospho-eNOS at Ser1177 [1:1000], anti-phospho-eNOS at Thr495 [1:1000], anti-eNOS [1:1000], anti-phospho-Src-kinase at Tyr416 [1:1000], anti-Src-kinase [1:1000], anti-phospho-caveolin-1 at Tyr14 [1:1000], anti-caveolin-1 [1:2000], and anti-β actin [1:10,000]) overnight at 4 °C. After incubation, the membranes were washed for 10 min thrice with TBST before being treated with horseradish peroxidase-conjugated anti-rabbit or anti-mouse IgG diluted to 1: 5000 for 60 min at 25 °C. The protein band’s signaling was stained with Westernbright^TM^ ECL western blotting detection kit (Advansta, Menlo Park, CA, USA) and imaged using a ChemiDoc^TM^ Touch Imaging System (Bio-Rad Laboratories Inc., Hercules, CA, USA). ImageJ software (version 1.45 s; National Institutes of Health, Bethesda, MD, USA) was used to quantify protein.

### 4.6. Measurement of Intracellular Calcium

A confocal laser microscope (IX70 Fluoview, Olympus, Tokyo, Japan) was used, as described previously, to measure intracellular calcium [Ca^2+^]_i_ level after amlodipine treatment [31]. HUVECs were seeded and cultured on a confocal cell culture dish (SPL, Pocheon, Republic of Korea). Cells were incubated with Fluo-4 AM (2.5 μM, Invitrogen, Waltham, MA, USA) in Hanks Balanced Salt Solution medium for 30 min, washed twice with phosphate-buffered saline solution, and treated with amlodipine (10^−6^ M) to measure intracellular calcium. HUVECs were pre-treated with 1% Intralipid or L-NAME, followed by amlodipine (10^−6^ M), 1% lipid emulsion, or L-NAME alone. Calcium levels were measured every 2.5 s at excitation and emission wavelengths of 485 and 520 nm, respectively. Intracellular calcium was analyzed using images expressed with Fluo-4 AM. Intracellular calcium was calculated as fluorescence intensity (F) divided by baseline fluorescence intensity (F_0_) before drug treatment. The net change in calcium ions was expressed as (F_max_-F_0_)/F_0_, where F_max_ is the maximum calcium level in fluorescence intensity after treatment with amlodipine, lipid emulsion, and L-NAME, either alone or combined. The intracellular calcium was measured for approximately 6 min.

### 4.7. Materials

All the high-purity chemicals are commercially available. Intralipid (20%) was purchased from Fresenius Kabi (Upsala, Sweden). Amlodipine, L-NAME, methylene blue, ODQ, phenylephrine, acetylcholine, linolenic acid, calmidazolium, and anti-β actin were obtained from Sigma-Aldrich (St Louis, MO, USA). Amlodipine, calmidazolium, and ODQ were dissolved in DMSO (final concentration of DMSO: 0.1%). Anti-Src-kinase, anti-phospho-Src-kinase, anti-phospho-eNOS (Ser 1177 and Thr 495), anti-caveolin-1, and anti-phospho-caveolin-1 antibodies were purchased from Cell Signaling Technology (Beverly, MA, USA). Anti-eNOS antibody was purchased from BD Bioscience (Franklin, NJ, USA).

### 4.8. Data Analysis

The study’s primary outcome was the effect of lipid emulsion, inhibitors, and linolenic acid (either alone or combined) on the amlodipine-induced vasodilation in the isolated rat aorta. This primary outcome was analyzed using a linear mixed effect model (Stata version 14.2, StataCorp LLC, Lakeway Drive, College Station, TX, USA) [32]. The effects of lipid emulsion, inhibitors, and amlodipine (either alone or combined) on the eNOS, Src-kinase, and caveolin-1 phosphorylation in HUVECs and cGMP formation in the isolated endothelium-intact aorta were analyzed using one-way analysis of variance, followed by Bonferroni’s multiple comparison tests (Prism 5.0, GraphPad Software, Inc., San Diego, CA, USA). The effects of lipid emulsion, L-NAME, and amlodipine (either alone or combined) on the intracellular calcium level in the HUVECs were analyzed using the Kruskal–Wallis test, followed by Dunn’s multiple comparison test. The difference in the area (min%) under phenylephrine-induced contraction (%) time (30–80 min) curves in the endothelium-intact aorta with or without L-NAME between groups with and without and 1% lipid emulsion post-treatment was calculated and analyzed using the unpaired Student’s *t*-test. *p* < 0.05 was considered statistically significant.

## 5. Conclusions

In conclusion, lipid emulsion attenuated endothelium-dependent nitric oxide-medi ated vasodilation induced using a toxic dose of amlodipine via inhibition of endothelial nitric oxide release, which seems to be mediated through reversal of stimulatory eNOS phosphorylation (Ser1177) and inhibitory eNOS (Thr495) dephosphorylation induced using amlodipine.

## Figures and Tables

**Figure 1 ijms-24-08741-f001:**
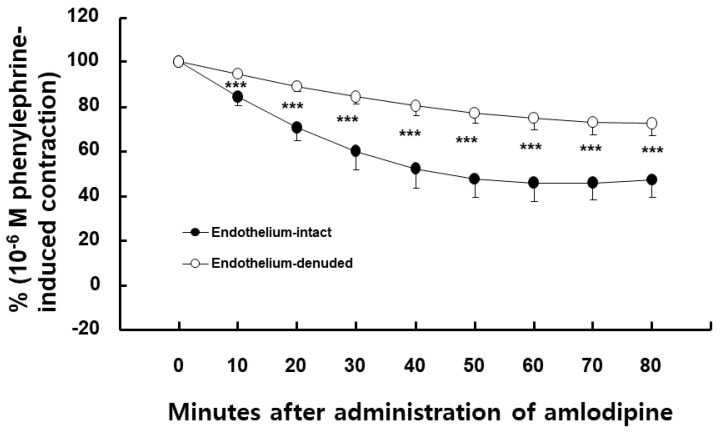
Effect of endothelial denudation on amlodipine (3 × 10^−7^ M)-induced vasodilation in isolated rat aorta. Data (N = 6) are presented as mean ± SD and expressed as percentage of phenylephrine-induced contraction. N indicates number of rats from which aortas were obtained. *** *p* < 0.001 vs. endothelium-intact.

**Figure 2 ijms-24-08741-f002:**
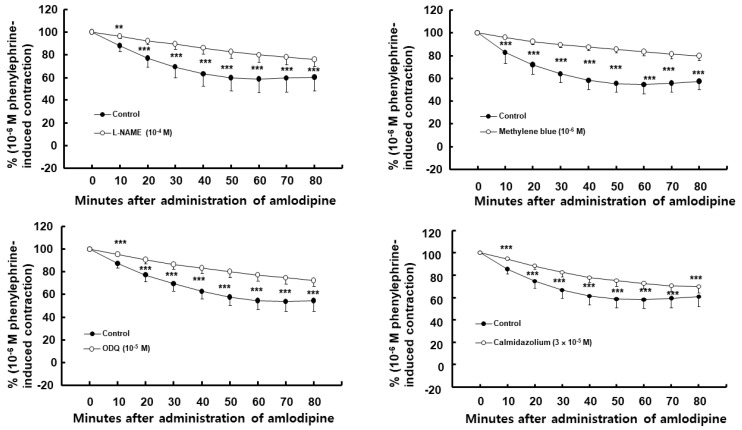
Effect of N^W^-nitro-L-arginine methyl (L-NAME, N = 7), methylene blue (N = 5), 1H-[1,2,4]oxadiazolo [4,3-a] quinoxalin-1-one (ODQ, N = 5), and calmidazolium (N = 6) on amlodipine (3 × 10^−7^ M)-induced vasodilation in endothelium-intact rat aorta. Data are presented as mean ± SD and expressed as percentage of phenylephrine-induced contraction. N indicates number of rats from which aortas were obtained. ** *p* < 0.01 and *** *p* < 0.001 vs. control.

**Figure 3 ijms-24-08741-f003:**
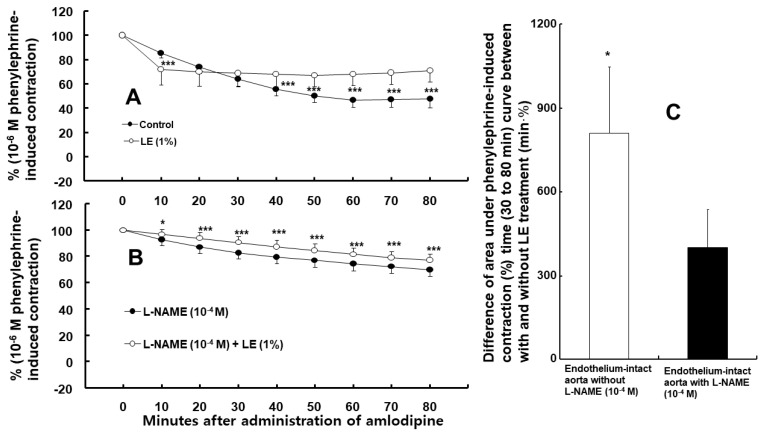
(**A**) Effect of post-treatment using lipid emulsion (LE, N = 5) on amlodipine (3 × 10^−7^ M)-induced vasodilation of isolated endothelium-intact aorta. Data are expressed as mean ± SD and expressed as percentage of phenylephrine-induced contraction. N indicates number of rats from which aortas were obtained. *** *p* < 0.001 vs. control. (**B**) Effect of post-treatment with LE (N = 5) on amlodipine (3 × 10^−7^ M)-induced vasodilation of isolated endothelium-intact aorta pre-treated with N^W^-nitro-L-arginine methyl (L-NAME). Data are presented as mean ± SD and expressed as percentage of phenylephrine-induced contraction. N indicates number of rats from which aortas were obtained. * *p* < 0.05 and *** *p* < 0.001 vs. L-NAME (10^−4^ M) alone. (**C**) Effect of L-NAME (10^−4^ M) on difference in area under phenylephrine-induced contraction (%) time (30–80 min) curves in endothelium-intact aorta between groups with and without LE treatment. Difference in area under phenylephrine-induced contraction time curves in endothelium-intact aorta is defined as area under phenylephrine-induced contraction (%) time (30–80 min) curves in LE-treated group minus area under phenylephrine-induced contraction (%) time (30–80 min) curves in groups without LE treatment. Data (N = 5) are presented as mean ± SD and expressed as difference in area (min%) under phenylephrine-induced contraction (%) time (30–80 min) curves. N indicates number of rats from which aortas were obtained. * *p* < 0.05 vs. endothelium-intact aorta with L-NAME (10^−4^ M).

**Figure 4 ijms-24-08741-f004:**
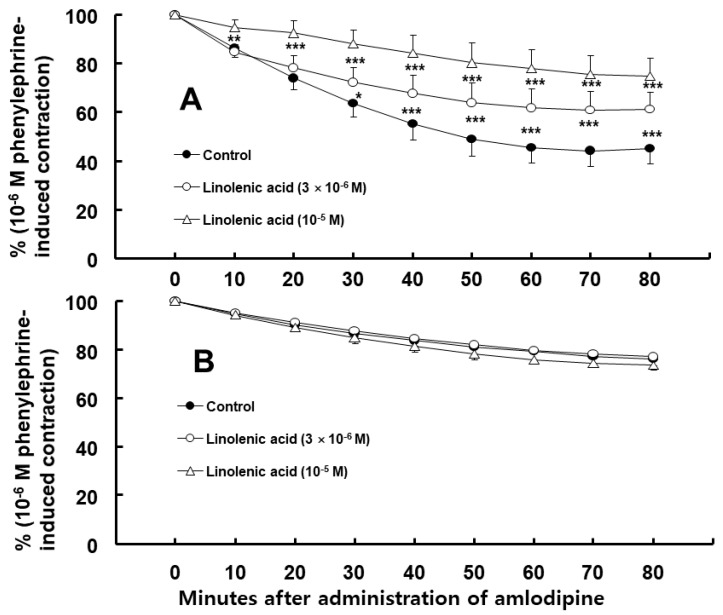
(**A**) Effect of post-treatment with linolenic acid (N = 8, 6, and 5 for control, and 3 × 10^−6^ and 10^−5^ M linolenic acid, respectively) on amlodipine (3 × 10^−7^ M)-induced vasodilation in endothelium-intact rat aorta. Data are presented as mean ± SD and expressed as percentage of phenylephrine-induced contraction. N indicates number of rats from which the aortas were obtained. ** p* < 0.05, ** *p* < 0.01 and *** *p* < 0.001 vs. control. (**B**) Effect of post-treatment with linolenic acid (N = 5) on amlodipine (3 × 10^−7^ M)-induced vasodilation in endothelium-denuded rat aorta. Data are presented as mean ± SD and expressed as percentage of phenylephrine-induced contraction. N indicates number of rats from which the aortas were obtained.

**Figure 5 ijms-24-08741-f005:**
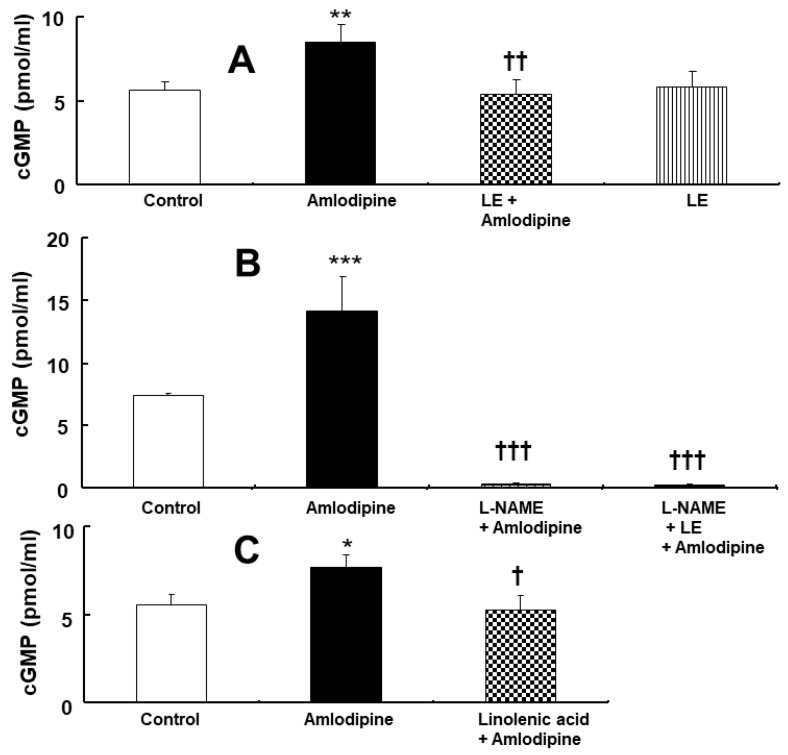
Effect of lipid emulsion (LE, 1%, **A**, N = 3), N^W^-nitro-L-arginine methyl (L-NAME, 10^−4^ M, **B**, N = 4), and linolenic acid (10^−5^ M, **C**, N = 3), either alone or in combination, on cyclic guanosine monophosphate (cGMP) formation induced via amlodipine (3 × 10^−7^ M) in isolated endothelium-intact aorta. Data are presented as mean ± SD. N indicates number of rats from which aortas were obtained. ** p* < 0.05, ** *p* < 0.01, and *** *p* < 0.001 vs. control. † *p* < 0.05, †† *p* < 0.01, and ††† *p* < 0.001 vs. amlodipine alone.

**Figure 6 ijms-24-08741-f006:**
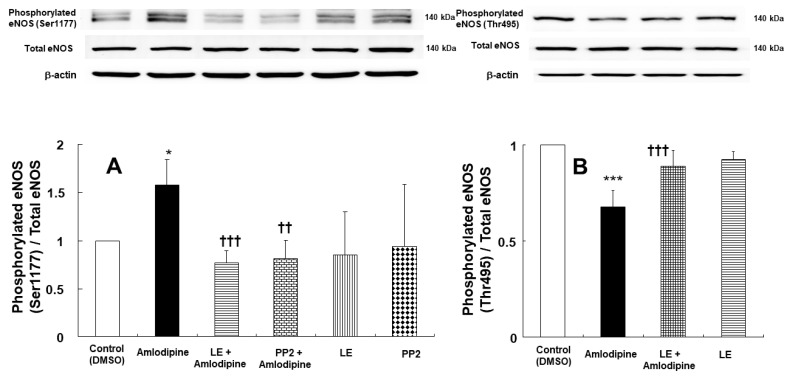
(**A**) Effect of lipid emulsion (LE, 1%), PP2 (2 × 10^−5^ M), and amlodipine (3 × 10^−7^ M), either alone or combined, on stimulatory endothelial nitric oxide synthase (eNOS, Ser1177) phosphorylation in human umbilical vein endothelial cells (HUVECs). Data (N = 5) are presented as mean ± SD. N indicates number of independent experiments. ** p* < 0.05 vs. control (DMSO). †† *p* < 0.01 and ††† *p* < 0.001 vs. amlodipine. (**B**) Effect of 1% LE and amlodipine, either alone or combined, on inhibitory eNOS (Thr495) phosphorylation in HUVECs. Data (N = 4) are presented as mean ± SD. N indicates number of independent experiments. *** *p* < 0.001 vs. control (DMSO). ††† *p* < 0.001 vs. amlodipine.

**Figure 7 ijms-24-08741-f007:**
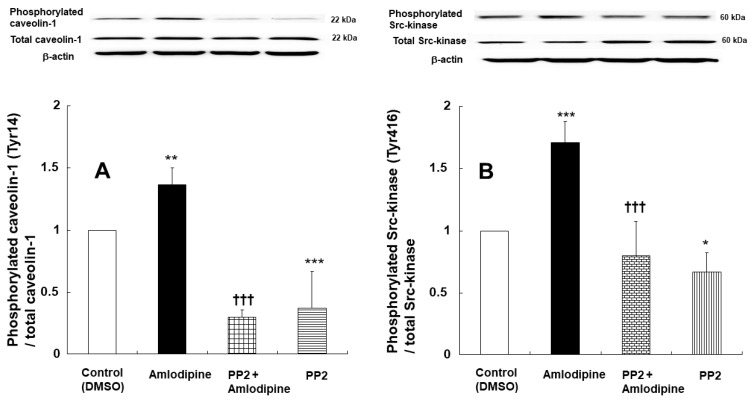
Effect of amlodipine (3 × 10^−7^ M) and PP2 (2 × 10^−5^ M), either alone or combined, on caveolin-1 (Tyr14, (**A**)) and Src-kinase (Tyr416, (**B**)) phosphorylation in human umbilical vein endothelial cells. Data (N = 4) are expressed as mean ± SD. N indicates number of independent experiments. * *p* < 0.05, ** *p* < 0.01, and *** *p* < 0.001 vs. control (DMSO). ††† *p* < 0.001 vs. amlodipine alone.

**Figure 8 ijms-24-08741-f008:**
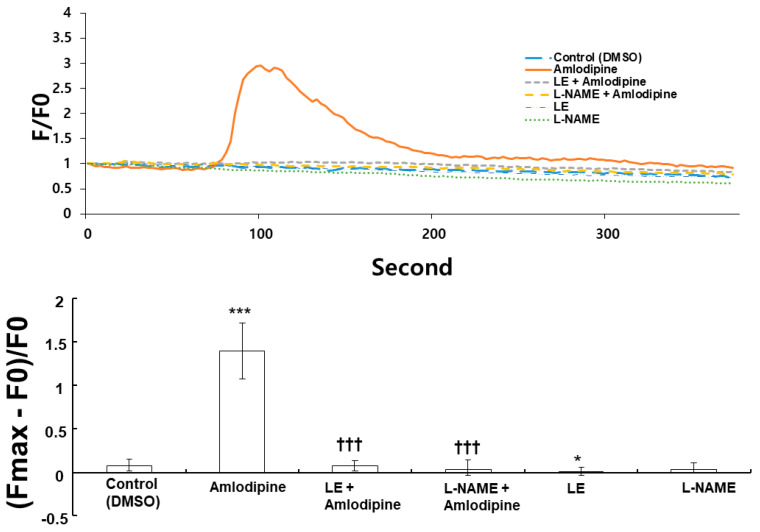
Effects of lipid emulsion (LE, 1%) and N^W^-nitro-L-arginine methyl (L-NAME, 10^−4^ M) on intracellular calcium level evoked via amlodipine (10^−6^ M) in human umbilical vein endothelial cells. Data (N = 5) are presented as median ± interquartile range (25–75%). N indicates number of independent experiments. * *p* < 0.05, and *** *p* < 0.001 vs. control (DMSO). ††† *p* < 0.001 vs. amlodipine alone.

## Data Availability

The data presented in this study are available on reasonable request from the corresponding author.

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
