# Peer review of "Lipid Emulsion Inhibits Amlodipine-Induced Nitric Oxide-Mediated Vasodilation in Isolated Rat Aorta"

_ijms, 2023, doi:10.3390/ijms24108741_

Round 1
Reviewer 1 Report
Comments and Suggestions for Authors
Park and colleagues have evaluated the effect of lipid emulsion on the vasodilation induced by a toxic dose of amlodipine in isolated rat aorta with a focus on nitric oxide. They found that lipid emulsion attenuated the vasodilation induced by amlodipine by inhibiting nitric oxide release.
The data are well presented and Results section is clear, but the text in Discussion part needs further clarification. Some points that need addressing in Discussion part:
1. line 232…Please provide explanation …on inhibition of vasodilation induced by amlodipine’s toxic dose seems to be mediated by linolenic acid-mediated inhibition of endothelial nitric oxide…
2. Please provide additional support of statement that amlodipine-induced eNOS phosphorylation (Ser1177) is mediated by Src-kinase-induced phosphorylation of caveolin-1.
3. Highlight the conclusions at the end of Discussion part.
Author Response
Response to Reviewer #1’s comments
Thank you very much for your valuable comments.
- line 232…Please provide explanation …on inhibition of vasodilation induced by amlodipine’s toxic dose seems to be mediated by linolenic acid-mediated inhibition of endothelial nitric oxide…
Response: We added the following modified sentence into the Discussion section of the revised manuscript. “Consistent with previous reports, linolenic acid-mediated inhibition of vasodilation induced by amlodipine’s toxic dose seems to be mediated by linolenic acid-mediated inhibition of endothelial nitric oxide by inhibiting eNOS, leading to the attenuation of vasodilation [20,21].”
- Please provide additional support of statement that amlodipine-induced eNOS phosphorylation (Ser1177) is mediated by Src-kinase-induced phosphorylation of caveolin-1.
Response: We added the following modified statement into the appropriate location in the Discussion section of the revised manuscript. “Stimulatory eNOS and caveolin-1 phosphorylation, induced by alpha-2 adrenoceptor agonist dexmedetomidine, was reportedly inhibited by Src-kinase inhibitor PP2 [24]. In addition, bupivacaine-induced inhibitory eNOS (Thr495) phosphorylation was mediated by Src-kinase and caveolin-1 pathway, attenuating the vasodilation [25]. Similar to previous reports, Src-kinase inhibitor PP2 inhibited stimulatory eNOS (Ser1177) and caveolin-1 (Tyr14) phosphorylation evoked by amlodipine and attenuated amlodipine-induced Src-kinase (Tyr416) phosphorylation [24]. Thus, considering previous reports, these results suggest that amlodipine-induced eNOS phosphorylation (Ser1177) is mediated by Src-kinase-induced phosphorylation of caveolin-1 [23].”
- Highlight the conclusions at the end of Discussion part.
Response: We added the following sentence at the end of the Discussion section in the revised manuscript. “In summary, lipid emulsion inhibited toxic doses of amlodipine-induced nitric oxide-mediated vasodilation, mediated by inhibiting nitric oxide production by reversing stimulatory and inhibitory eNOS phosphorylation.”

Reviewer 2 Report
Dear Park et al.,
Thank you for submitting your manuscript entitled “Inhibitory Effect Of Lipid Emulsion On The Amlodipine-in-2 duced Nitric Oxide-mediated Vasodilation” to “International Journal of Molecular Sciences”. After carefully reviewing your submission, I have the following comments and recommendations:
Major concern:
· I recommend providing a convincing justification for focusing on Lipid Emulsion than L-NAME, methylene blue, ODQ, calmidazolium, and linolenic acid in abstract, introduction and discussion section. Otherwise, the title of the manuscript is quite confusing.
· It is quite unclear the importance of using L-NAME, methylene blue, ODQ, calmidazolium, and linolenic acid in this study?
· Why did you use HUVECs than rat aortic endothelial cells for figure 6?
· Why did you not use one-way anova for statistical analysis of figure 8?
· The figure of intact and denuded endothelium is missing. Any reference for preparation of rat aorta?
· Why has the study done on ex-vivo setup? Why in vivo method has not used?
Minor concern:
· It would be better to describe the results of each figure under separate subheadings than describing combindly, which might be interesting and easier to follow to the readers.
· Sentences between 130-137 are quite unclear. Please rewrite.
· Limitation of the study has not been described, which could make the manuscript more acceptable.
·
· T cell subsets, combining with PNI on discussion section. Otherwise, the manuscript's significance will be diluted.
· On line 98-100, the mentioned cut-off points using ROC were not shown on figures. How did you get the values need proper description and figures, otherwise it’s confusing to the readers. On line 14-15, “………evaluated the patient’s status….” will be “………evaluated the patients’ status ……..” as you are dealing with many patients, not one.
Overall, I find the manuscript interesting, and I believe it has the potential to make a valuable contribution to the field. However, I recommend making the revisions outlined above before the manuscript can be considered for publication.
Thank you for considering my comments and recommendations, and I look forward to seeing a revised version of your manuscript.
The quality of English language is quite acceptable.
Author Response
Response to Reviewer #2’s comments
Thank you very much for your valuable comments.
Major concern:
- I recommend providing a convincing justification for focusing on Lipid Emulsion than L-NAME, methylene blue, ODQ, calmidazolium, and linolenic acid in abstract, introduction and discussion section. Otherwise, the title of the manuscript is quite confusing.
Response: We changed the title to “Lipid Emulsion Inhibits Amlodipine-induced Nitric Oxide-mediated Vasodilation” in the revised manuscript. In addition, we deleted the comment regarding the ODQ and calmidazolium in the Abstract. We added the following sentences to the Abstract in the revised manuscript. “Lipid emulsion inhibited amlodipine-induced endothelial intracellular calcium increase.”
- It is quite unclear the importance of using L-NAME, methylene blue, ODQ, calmidazolium, and linolenic acid in this study?
Response: Thank you for your thoughtful comments. First, as we would like to confirm whether amlodipine-induced vasodilation is mediated by endothelial nitric oxide, the effects of endothelial denudation and inhibitors (L-NAME, ODQ, methylene blue, and calmidazolium) of the cellular signal pathway, associated with nitric oxide-mediated vasodilation, on the amlodipine-induced vasodilation in isolated rat aorta were examined. After we confirmed that amlodipine-induced vasodilation was mediated by endothelial nitric oxide, we examined the effect of lipid emulsion, linolenic acid, which is one of the long-chain fatty acids contained in Intralipid, and L-NAME, alone or combined, on the amlodipine-induced vasodilation in the endothelium-intact aorta and underlying mechanism with particular focus on endothelial nitric oxide.
- Why did you use HUVECs than rat aortic endothelial cells for figure 6?
Response: As I mentioned in the Discussion section, this is a limitation of this study. We have been trying to perform the primary culture of rat aortic endothelial cells recently. We will use this next time after we succeed in the primary culture of rat aortic endothelial cells. We added the following sentences regarding the disadvantage of HUVECs into the Discussion section of the revised manuscript. “HUVECs have disadvantages, such as sex differences and some degree of heterogeneity [29].”
- Why did you not use one-way anova for statistical analysis of figure 8?
Response: As the relevant data did not pass the normality test, we used the Kruskal–Wallis test, followed by Dunn’s multiple comparison test.
- The figure of intact and denuded endothelium is missing. Any reference for preparation of rat aorta?
Response:
Please refer to Figure S2: Original tracing showing acetylcholine-induced vasodilation in endothelium-intact and -denuded rat aorta precontracted using phenylephrine.
In addition, please refer to the following sentences and references in the Preparation of rat aorta and isometric tension measurement section. “The isolated rat aorta for assessing isometric tension was prepared, as previously described [20].” “The endothelial integrity of endothelium-intact rat aorta was examined as follows [20].” “The endothelial denudation was verified as follows [20].”
- Why has the study done on ex-vivo setup? Why in vivo method has not used?
Response: As this study examined the underlying mechanism responsible for lipid emulsion-mediated reversal of amlodipine-induced nitric oxide-mediated vasodilation, first, we experimented using isolated rat aorta. After we confirm lipid emulsion-mediated inhibition of amlodipine-induced nitric oxide-mediated vasodilation via inhibition of nitric oxide production, we will try a further in vivo study comparing the effect of lipid emulsion and non-specific GC inhibitor methylene on the severe hypotension induced by a toxic dose of amlodipine. As I mentioned in the Discussion section, this is one of the limitations of this in vitro study. We added the following sentence to the Discussion section of the revised manuscript. “Third, an in vivo study is more appropriate to examine the effects of lipid emulsion on cardiovascular depression caused by amlodipine’s toxic dose. However, our study was an in vitro study. Therefore, a further study comparing the effect of lipid emulsion and non-specific GC inhibitor methylene blue on hypotension induced by a toxic dose of amlodipine in an in vivo model is required to confirm this result.”
Minor concern:
- It would be better to describe the results of each figure under separate subheadings than describing combindly, which might be interesting and easier to follow to the readers.
Response: We added four subheadings to the Results section of the revised manuscript.
- Sentences between 130-137 are quite unclear. Please rewrite.
Response: We slightly modified the relevant sentence. The revised sentence is as follows: “However, the lipid emulsion (1%) did not affect the amlodipine (3 ´ 10-7 M)-induced cGMP formation in the endothelium-intact rat aorta pretreated with L-NAME (10-4 M) (Figure 5B).”
- Limitation of the study has not been described, which could make the manuscript more acceptable.
Response: The modified and added sentences described in the Discussion section of the revised manuscript are as follows: “This study has a few limitations. First, the peripheral vascular resistance, which contributes to blood pressure, is mainly determined by a small resistance arteriole. However, in this study, the rat aorta was regarded as a conduit vessel [28]. Second, for isometric tension measurement, we used isolated rat aorta. However, for the western blot, we used HUVECs. HUVECs have disadvantages, such as sex differences and some degree of heterogeneity [29]. Third, an in vivo study is more appropriate to examine the effects of lipid emulsion on cardiovascular depression caused by amlodipine’s toxic dose. However, our study was an in vitro study. Therefore, a further study comparing the effect of lipid emulsion and non-specific GC inhibitor methylene blue on hypotension induced by a toxic dose of amlodipine in an in vivo model is required to confirm this result.”
- T cell subsets, combining with PNI on discussion section. Otherwise, the manuscript's significance will be diluted.
Response: This comment is not related to this manuscript.
- On line 98-100, the mentioned cut-off points using ROC were not shown on figures. How did you get the values need proper description and figures, otherwise it’s confusing to the readers. On line 14-15, “………evaluated the patient’s status….” will be “………evaluated the patients’ status ……..” as you are dealing with many patients, not one.
Response: This comment is not related to this manuscript.

Round 2
Reviewer 2 Report
Dear Park et al.,
Thanks for the revised version of the manuscript. I would prefer to mention clearly on the title and abstract that the study has done in isolated rat aorta.
Author Response
- Thanks for the revised version of the manuscript. I would prefer to mention clearly on the title and abstract that the study has done in isolated rat aorta.
Response: We changed the title to “Lipid Emulsion Inhibits Amlodipine-induced Nitric Oxide-mediated Vasodilation in Isolated Rat Aorta” in the revised manuscript. In addition, we added the following words “in isolated rat aorta” to the Abstract in the revised manuscript.
